# Co-Treatment with Single and Ternary Mixture Gas of Dimethyl Sulfide, Propanethiol, and Toluene by a Macrokinetic Analysis in a Biotrickling Filter Seeded with *Alcaligenes* sp. SY1 and *Pseudomonas Putida* S1

Yiming Sun [1,2,3,†], Xiaowei Lin [1,†], Shaodong Zhu [4], Jianmeng Chen [5], Yi He [1,3], Yao Shi [1,2,*], Hua Liu [6] and Lei Qin [5]

[1]  College of Chemical and Biological Engineering, Zhejiang University, Hangzhou 310027, China; sunyiming@kaitiangroup.com (Y.S.); linxw@zju.edu.cn (X.L.); yihezj@zju.edu.cn (Y.H.)

[2]  Key Laboratory of Biomass Chemical Engineering of Ministry of Education, Zhejiang University, Hangzhou 310027, China

[3]  Department of Chemical Engineering, University of Washington, Seattle, WA 98195, USA

[4]  Shaoxing Environmental Monitoring Central Station, Shaoxing 312000, China; zuestone@dingtalk.com

[5]  College of Biological and Environmental Engineering, Zhejiang University of Technology, Hangzhou 310014, China; jchen@zjut.edu.cn (J.C.); qinlei0214@zjut.edu.cn (L.Q.)

[6]  China Aerospace Kaitian Environmental Technology Co., Ltd., Changsha 410100, China; liuh@kaitiangroup.com

*  Correspondence: shiyao@zju.edu.cn; Tel.: +86-13336085801; Fax: +86-571-88273591

†  These authors contributed equally to this work.

**Abstract:** The biotrickling filter (BTF) treatment is an effective way of dealing with air pollution caused by volatile organic compounds (VOCs). However, this approach is typically used for single VOCs treatment but not for the mixtures of VOC and volatile organic sulfur compounds (VOSCs), even if they are often encountered in industrial applications. Therefore, we investigated the performance of BTF for single and ternary mixture gas of dimethyl sulfide (DMS), propanethiol, and toluene, respectively. Results showed that the co-treatment enhanced the removal efficiency of toluene, but not of dimethyl sulfide or propanethiol. Maximum removal rates ($r_{max}$) of DMS, propanethiol and toluene were calculated to be 256.41 g·m$^{-3}$·h$^{-1}$, 204.08 g·m$^{-3}$·h$^{-1}$ and 90.91 g·m$^{-3}$·h$^{-1}$, respectively. For a gas mixture of these three constituents, $r_{max}$ was measured to be 114.94 g·m$^{-3}$·h$^{-1}$, 104.17 g·m$^{-3}$·h$^{-1}$ and 99.01 g·m$^{-3}$·h$^{-1}$, separately. Illumina MiSeq sequencing analysis further indicated that Proteobacteria and Bacteroidetes were the major bacterial groups in BTF packing materials. A shift of bacterial community structure was observed during the biodegradation process.

**Keywords:** biotrickling filter; decomposition; dimethyl sulfide; pollution; propanethiol; toluene

## 1. Introduction

The removal of odorous waste, including volatile organic sulfur compounds (VOSCs) has been paid increasing attention for air pollution control solutions, considering their aqueous solubility and gradual diffusion into the atmosphere [1–3]. VOSCs are extremely toxic, corrosive, and malodorous compounds, which have significant effects on animal and human health as well as the environment [4]. Moreover, VOSCs very often co-exist with volatile organic compounds (VOCs). BTEX, including benzene, toluene, ethylbenzene, and xylene as paint thinner, are widely applied in machining operations and plating shops. The mixture of dimethyl sulfide (DMS), propanethiol, and toluene can be found in the exhaust gas from pharmaceutical, pesticide, and rubber industries. For example, an exhaust gas from a pesticide company can contain DMS (C DMS = 100 mg·m$^{-3}$~300 mg·m$^{-3}$), propanethiol (C Propanethiol = 30 mg·m$^{-3}$~150 mg·m$^{-3}$), and toluene (C Toluene = 100 mg·m$^{-3}$~400 mg·m$^{-3}$). Thus, it is worthwhile to focus on

some typical VOSCs and VOCs, such as DMS, propanethiol, and toluene. As a result, the development of highly efficient, economically feasible, and environmentally friendly technologies of odor removal is in urgent demand.

The traditional approaches for the removal of VOSCs and VOCs from a gas stream are mostly physical methods (filtration, adsorption, dilution by clean air, and condensation) and chemical methods (chemical oxidation, thermal oxidation, chemical adsorption, and UV activation method) [5–7]. Recently, these methods have been used as a pretreatment, in terms of their removal mechanism. The transformation of pollutants from the gas phase to other phases or breaking down long chemical chains to short chemical chains is not a complete removal of pollutants. Although chemical solutions are efficient for gaseous pollutant removal, some of them are expensive, non-recyclable, or introduce secondary pollution, even for a new green chemical called ionic liquid [8]. Moreover, the waste gas in the pharmaceutical, pesticide, and rubber industries not only contains VOCs and VOSCs, it also contains dichloromethane. BTF compares with the other conventional approaches, such as regenerative thermal oxidization (RTO) or regenerative catalytic oxidation (RCO), which have their own limitations, including the chlorine being corrosive against metal and the operation cost of RTO or RCO is expensive. BTF not only transfers VOCs or VOSCs to $CO_2$ and $H_2O$, it also transfers to the intermediates and biomass in an eco-friendly way. (4) Comparing BTF with other popular approaches, such as RTO or RCO, shows that biotrickling will not generate secondary air pollutants, such NOx. In addition, it also reduces the $CO_2$ content in products.

In comparison, biopurification (biofilters, bioscrubbers, biotrickling filters) is a practical and preferred approach for the treatment of VOSCs and VOCs [9–12], and the key difference between biotrickling filters and bioscrubbers are that they target different application scenarios in the field. Biotrickling filters are suitable for factories treated with a mixture of VOCs, and bioscrubbers are suitable for centralized wastewater treatment scenarios treated with hydrophilic VOCs. Among these various types of biological treatment, biotrickling filters (BTFs) are preferred in many cases due to their stable operation, low-cost, and easy processing control [13–18]. Therefore, biotrickling filter has an advantage over many biological treatment technologies in the control of operating parameters and mineralized efficiency, especially for high concentration acidified contaminants containing waste gas streams, such as sulfur, chlorine, or nitrogen containing compounds [19–21]. It has been reported to be effective for treating single VOSC, such as methanethiol or dimethyl sulfide [22,23]. Few papers have been published using an aerobic biotrickling filter system for removal of waste gas containing ethanethiol [10,24,25] besides our previous paper [26]. And few studies have been conducted to purify waste gas containing DMDS (dimethyl disulfide) via biofilter [27–31]. Little research has been reported to get thioanisole waste gas biologically treated. It is worthwhile pointing out that various odorous pollutants always co-exist in a real environment [32,33]. Nevertheless, in strong contrast to the level of understanding towards the degradation of single organic compound, little is known regarding the biotreatment of waste gas mixture, especially the binary mixture gas of ethanethiol and DMDS and ternary mixture gas of ethanethiol, DMDS, and thioanisole, which are more often encountered in practical applications.

In previous studies about biopurification for gaseous pollutants, various types of single VOSC or VOC have received extensive attention [34–37], but very little attention has been paid to control mixtures of VOSC and VOC from industries. Furthermore, very few have been understood towards the co-treatment of VOSCs and VOCs, especially the mixture gas which is very often found in real cases. It has been reported that a single substrate, such as DMS, propanethiol, or toluene can be removed efficiently from BTFs [29,38]. Although, few papers have been published using an aerobic BTF to remove VOCs and VOSCs together [22,23]. Few papers have investigated single pollutant performance co-existing with others [39,40]. However, the mechanism for how the mixture of contaminants affects the kinetics of the individual remains unclear. Therefore, it is worth focusing on the co-

treatment of typical VOSCs and VOCs in terms of those contaminants, which are common in reality.

In this work, a BTF, inoculated with the DMS isolator *Alcaligenes* sp. SY1 [23], propanethiol isolator *Pseudomonas putida* S1 [41], and the acclimated activated sludge with toluene degradability, was set up for the treatment of the mixture containing DMS, propanethiol, and toluene. Furthermore, the biofilms from three different periods were collected and sequenced using the Illumina MiSeq platform for a comprehensive understanding of microbial compositions and structures.

## 2. Experimental Methods

### 2.1. Microorganism Culture and Medium Preparation

The microorganism seeding method and the degradation ability for DMS and propanethiol of *Alcaligenes* sp. SY1 and *Pseudomonas putida S1*, were the same methods utilized in our previous research [26,41]. A few acclimatized activated sludges were also used in system.

A mineral salt medium (MSM) was contained: 4.5 g $Na_2HPO_4 \cdot 12H_2O$, 2.5 g $KH_2PO_4$, 1.0 g $NH_4HCO_3$, 0.2 g $MgCl_2 \cdot 7H_2O$, 0.03 g $CaCl_2 \cdot 2H_2O$, 0.01 g $FeCl_2 \cdot 4H_2O$ in 1 L deionized water and 1 mL of trace element stock solution. The trace element stock solution contained 1.0 $g \cdot L^{-1}$ $FeCl_2 \cdot 7H_2O$, 0.02 $g \cdot L^{-1}$ $CuCl_2 \cdot 5H_2O$, 0.014 $g \cdot L^{-1}$ $H_3BO_3$, 0.10 $g \cdot L^{-1}$ $MnCl_2 \cdot 4H_2O$, 0.10 $g \cdot L^{-1}$ $ZnCl_2$ $7H_2O$, 0.02 $g \cdot L^{-1}$ $Na_2MoO_4 \cdot 2H_2O$, and 0.02 $g \cdot L^{-1}$ $CoCl_2 \cdot 6H_2O$ in a 250 mL flask and the flask was sealed with a rubber stopper [38]. 12 L DMS and propanethiol was injected into each flask and the total volume of solution was 500 mL for each bacteria strain. After being cultured at 30 °C and 160 r/min, suspension containing a specific bacteria strain was obtained as the inoculant for the biofilter.

### 2.2. BTF Setup and Operation Condition

The schematic of a multi-layer BTF is shown in Supplementary Materials Figure S1; the BTF was constructed using a plexi-glass with a total height of 90 cm, an inner diameter 12 cm, an outer diameter of 15 cm, and the height of 25 cm for each layer. The BTF was packed with a polyurethane pall ring, which has an initial porosity 0.91. Gas sampling ports were located along the height of the biofilter, with another two sampling ports located in each layer for the collection of biomass samples.

The air stream for the BTF was built from two sub air streams. One air stream carries pollutants by passing air through a glass bubbler of 250 mL containing liquid VOCs at room temperature and then mixed with major air stream in mixture reactor in order to homogeneous mixing trough BTF reactor. Concentrations of VOCs in the synthetic gas stream were varied by changing the air flow rate through the bubbler. The pH of the system was adjusted by HCl (10%) solution through an auto-pH adjustor (Kesheng, Hangzhou, China) and was controlled in the range of 6.86 to 7.20. Pressure drop in the gas phase through the fixed bed reactor was measured by a manometer (AZ8252, Guangzhou, China). The nutrient solution was sprayed over the top of the column at 6 $L \cdot h^{-1}$ by a peristaltic pump (Longerpump® BT600-2J, Hangzhou, China). Gas samples were collected by gas tight syringes (Gaoge, Shanghai, China) fitted with 3-way luer-lock and measured by GC analysis. The measurement procedure for weight of packing material, porosity, and liquid maintaining capacity was based on our previous work [32].

During the startup period, BTFs were operated in a continuous manner, i.e., the nutrient liquid trickled through the packing for 24 h. The trickling rate was controlled at 6 $L \cdot h^{-1}$. 50% of the trickling liquid was replaced every 5 days. The reactor was operated in a closed loop mode with respect to liquid to maximize cell adhesion to the packing material. The start-up method of the BTF was the method reported by Jin [34]. The BTF was initially started up with synthesized gas flow and the corresponding empty bed retention time (EBRT) was 56 s. Initial concentrations were 200.0 $mg \cdot m^{-3}$ for DMS and 100.0 $mg \cdot m^{-3}$ for propanethiol.

### 2.3. Analytical Method

DMS, propanethiol, and toluene were measured by gas chromatography (SHIMADZU GC-14B, Kyoto, Japan,) coupled with flame photometric detector (FPD, SHIMADZU, Japan) and flame ionization detector (FID, SHIMADZU, Japan). Hydrogen and air flow rates for FPD and FID were 50 mL·min$^{-1}$ and nitrogen used as carrier gas was 200 mL·min$^{-1}$. Temperatures of the injection, oven, and detection were 180 °C, 100 °C, and 180 °C, respectively.

The $CO_2$ content was determined through GC9790 gas chromatograph (Fuli TDX-01, 30 m × 0.32 mm × 20μm, Technologies, Taizhou, China). The pH value of the sample was adjusted to 2.0, making sure the inorganic $HCO_3^-$ transfer to $CO_2$ completely. The detected samples were collected by the microsampler from headspace vapors of closed system and analyzed by using thermal conductivity detector (TCD). The temperature of the oven and detection were 100 °C and 150 °C, separately. The column flow rate was set as 5 mL·min$^{-1}$. The external standard method was utilized to calculate $CO_2$ concentration.

The startup period of BTF was from days 0–days 20, the steady period was from days 21–days 140, the end period was from days 141–days 252.

### 2.4. Biofilm Formation and Microbial Analyses

DMS (J&K®, Beijing, China), propanethiol (J&K®, Beijing, China) and toluene (Aladdin®, Shanghai, China) were utilized in this study. All other chemicals were high performance liquid chromatography (HPLC) grade. Biofilm formation of samples on day 20, 77 and 240 were observed by using Field Emission Scanning Electron Microscope (FESEM) (HITACHI, SU8010, Kyoto, Japan,). The samples were sent to Sangon Biotech Co., Ltd., (Shanghai, China) for high-throughput sequencing. DNA samples were amplified using the primers 515F (5′-GTGCCAGCMGCCGCGG-3′), and 806R (5′-GGACTACHVGGGTWTCTAAT-3′), targeting the V4 hypervariable regions of 16S rRNA genes [37,42,43].

Operational taxonomic units (OTUs) were assigned by applying UPARSE, and with 97% of sequences similarity were clustered into OTUs. Each OTU was assigned by using the Ribosomal Database Project (RDP) classifier to identify the typical sequence [44].

### 2.5. Operation Parameters of BTF

The inlet loading rate (ILR) of BTF is defined as,

$$ILR = \left(\frac{C_i}{V}\right) \times Q \tag{1}$$

where $C_i$ and $Q$ are the mass concentration of the contaminants (%) and the mass flow rate of the inlet stream (g·h$^{-1}$), respectively. $V$ is the volume of BTF bioreactor (m$^3$).

The performance of the BTF is evaluated by eliminate capacity (EC) (g·m$^{-3}$·h$^{-1}$), EBRTs, and RE (%). EC is the measurement of contaminant removal capacity at a given ILR. EBRT is the measurement of gas residence time within the reactor medium. ILR, EC, EBRT, and RE were calculated using the following equations:

Eliminate Capacity:

$$EC = (\frac{C_i - C_o}{V}) \times Q \tag{2}$$

Empty bed resistance time:

$$EBRT = \frac{V}{Q} \tag{3}$$

Removal efficiency:

$$RE = (\frac{C_i - C_o}{C_i}) \times 100 \tag{4}$$

### 2.6. BTFs Operation and Startup Period

BTFs operated for 252 days to treat individual and synthesis gas. The study was performed for individual and ternary co-treatment with various parameters, such as inlet

loading rate (ILR) and empty bed retention time (EBRT). The ILR and EBRT experiment conditions changes are presented in detail in Table S1.

During the startup period (day 1 to day 22), the inlet concentration of the mixture pollutants was maintained around 200.0 mg·m$^{-3}$ at the ratio of DMS, propanethiol and toluene at 2:1:1 with a gas flow rate of 0.2 m$^3$·h$^{-1}$. In other words, the ILR and EBRT were maintained at 11.4 g·m$^{-3}$·h$^{-1}$ and 56 s, respectively. Over 80.0% of removal efficiency (RE) was observed in the BTF treating DMS on day 20, while the removal efficiency of propanethiol and toluene was stable after 14 and 19 days, respectively. The system was seeded with *Alcaligenes* sp. SY1 and *Pseudomonas putida* S1.

### 2.7. Kinetic Analysis

At the steady state, the growth rate of biofilms was equaled by their decay rate, the biological system was balanced. Therefore, a Michaelis–Menten (M–M) model can be applied as a macrokinetic analysis to the BTF system, which in terms of kinetic constants, was stable during the period. The degradation rate of contaminants within the biofilms was investigated in the gas-phase BTF and could be estimated by following equation:

$$\frac{V}{Q(C_i - C_o)} = \frac{K_s}{r_{max}C_{ln}} + \frac{1}{r_{max}} \tag{5}$$

in which $r_{max}$ is the maximum biodegradation rate (g·m$^{-3}$·h$^{-1}$) and $K_s$ is the saturation M–M model constant (mg·L$^{-1}$) in the gas phase. $C_{ln}$ is the natural logarithm mean concentration [$(C_i - C_o)/\ln(C_i/C_o)$]. According to the linear relationship between $1/C_{ln}$ and [$V/Q(C_i - C_o)$] could be calculated from the intercept and slope, respectively [45,46].

## 3. Results and Discussion

### 3.1. Performance of BTF for Mixed Gas during Different Phases

After the startup period, the BTF was tested under different conditions from phase I to phase III. The variations of inlet and outlet concentration of mixture pollutants in different phases are shown in Figure 1. EBRTs of 56 s, 28 s and 20 s were designed for phase I to phase III.

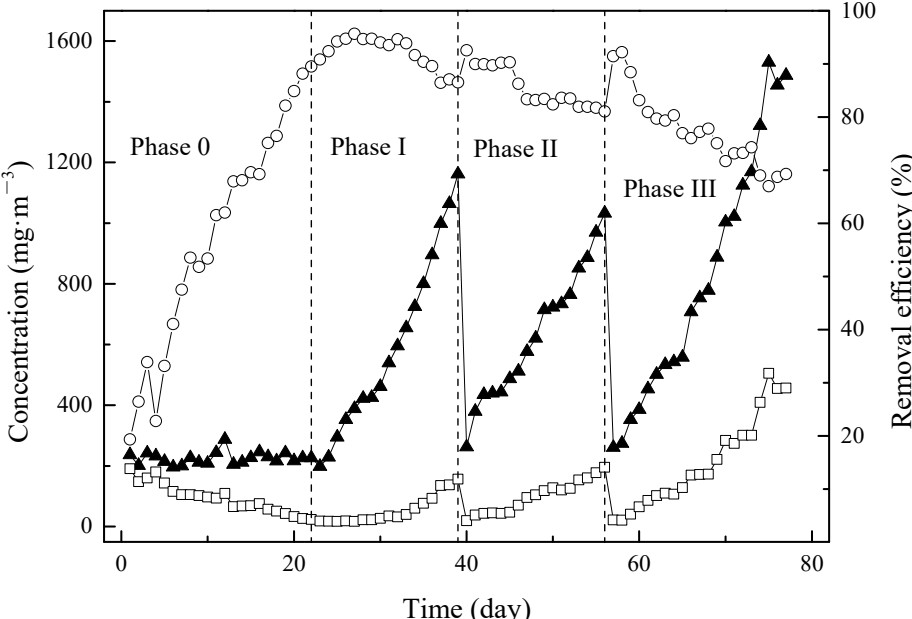

**Figure 1.** Inlet (▲) and outlet (□) concentrations corresponding with removal efficiency (○) of mixture contaminants in BTF during different phases. (0): startup period EBRT 28 s (I): EBRT 56 s (II): EBRT 28 s (III): EBRT 20 s.

During phase I (from day 23 to day 39), initially, RE was increased with the growth of inlet concentration. However, due to sudden enhancement of inlet concentration to 998.67 mg·m$^{-3}$ from 895.64 mg·m$^{-3}$ on day 37, the RE was decreased from 89.56% to 86.45% and maintained constantly until the end of phase I. The similar tendency was observed during phase II (day 40 to day 56), RE was slightly increased with an enhancement of inlet concentration until day 46. When the inlet concentration was enhanced from increased with the growth of inlet concentration. However, due to sudden enhancement of inlet concentration to 621.07 mg·m$^{-3}$ to 714.70 mg·m$^{-3}$, RE was suddenly dropped from 90.30% to 86.20% and keep gradually decreased to the end of phase II at 81.10%. During the phase III (day 57 to day 77), the tendency was different. An obvious outlet concentration increase was observed during phase III, while no significant outlet concentration enhancement was observed during phase I and phase II. Moreover, RE drops from 88.41% to 80.90% without an obvious increasing period during phase III.

### 3.1.1. Effect of ILR (Inlet Loading Rate) on the EC of Mixture Treatment

The different EC corresponding to ILR variations for the mixed pollutants treatment is presented in Figure 2. The behavior of BTF in mixture pollutants treatment also can be categorized into two parts according to liner regression: (I) diffusion constraints (DC, up to 130.00 g·m$^{-3}$·h$^{-1}$ of ILR according to Figure 2); and (II) reaction constraints (RC, ILRs greater than 130.00 g·m$^{-3}$·h$^{-1}$). Diffusion rate and reaction rate both significantly affected the total behavior of BTF during the operation [47–49]. In DC, reaction rate in the biofilm was greater than the diffusion rate from the gaseous phase to aqueous phase, and as a result, diffusion rate was the limiting factor, in terms of a lower ILR. Contrarily, with a higher ILR, the diffusion rate was greater than the reaction rate; the resulting reaction rate was to be the control factor. Thus, it could be due to the restriction of reaction rate, data slightly deviated from line of 100% of EC when ILR grew higher.

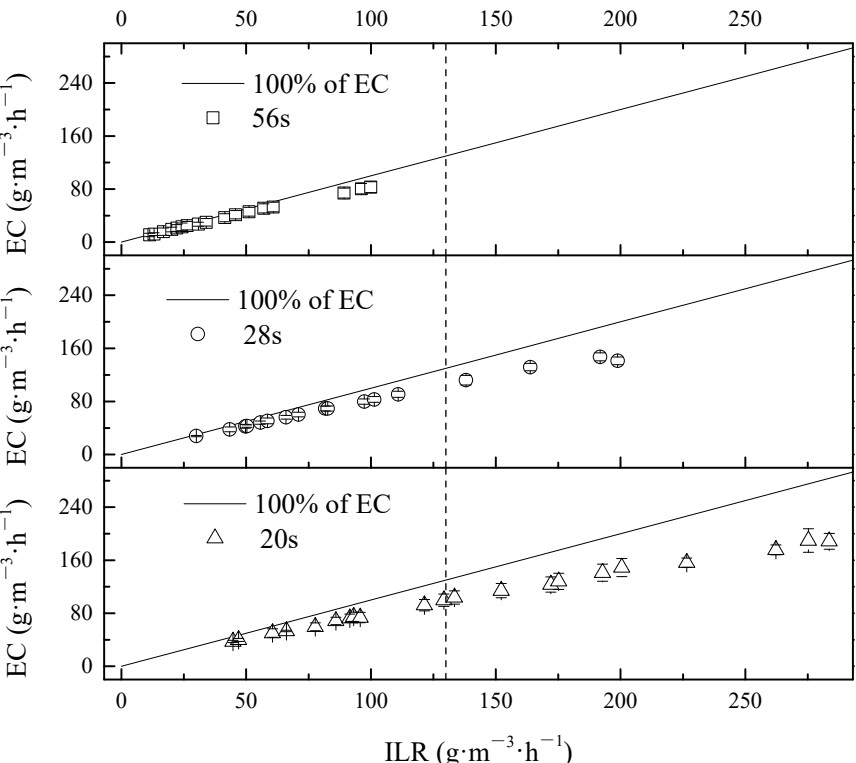

**Figure 2.** Effect of ILR on RE of BTF in mixture treatment during different phases.

The variation of EC with different ILR under different EBRTs for DMS, propanethiol, and toluene in mixture treatment by BTF is presented in Figure 3a–c. At high ILR, ECs of DMS did not deviate far from the ideal line of the 100% removal line. The tendency

of propanethiol and toluene were similar. As shown in Figure 3a–c, the maximum EC of DMS and propanethiol were observed on maximum ILR under EBRT of 20 s; however, the maximum EC of toluene occurred under EBRT of 28 s. It confirmed that marginal toluene ILR of RC was lower than the other two contaminants.

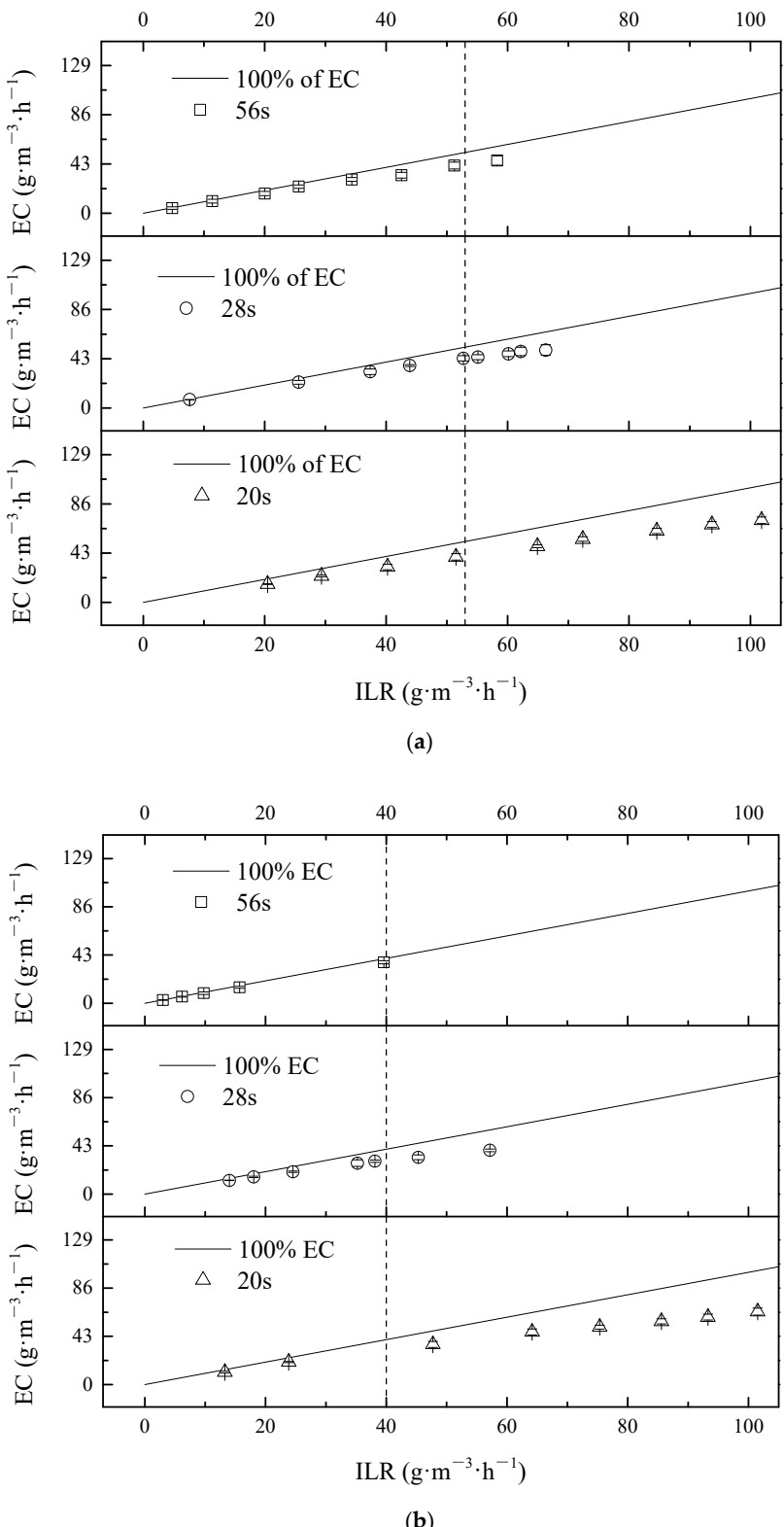

**Figure 3.** *Cont.*

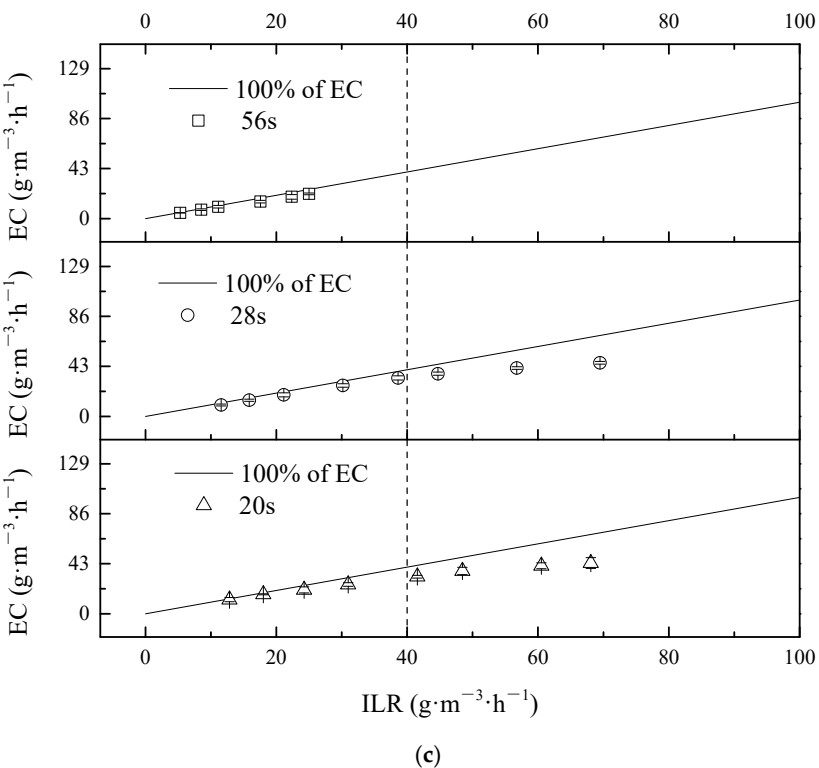

**Figure 3.** Effect of ILR on EC of individual contaminants: (**a**) DMS; (**b**) propanethiol and (**c**) toluene in mixture treatment during different phases.

Figures 2 and 3 show that the total maximum EC of overall contaminants were achieved at 80.70 $g·m^{-3}·h^{-1}$ (DMS 50.4%, propanethiol 31.3%, toluene 18.2%), 145.91 $g·m^{-3}·h^{-1}$ (DMS 37%, propanethiol 28.9%, toluene 34.1%) and 189.70 $g·m^{-3}·h^{-1}$ (DMS 40.0%, propanethiol 36.3%, toluene 24.0%) at different EBRTs of 56 s, 28 s, and 20 s, respectively. Among these pollutants, the value of ILR in terms of certain EC followed the order: DMS > propanethiol > toluene; DMS was thus the most stubborn chemical among these three in treatment.

### 3.1.2. Effect of ILR on the RE

According to Figure 2, total RE of mixture contaminants was maintained above 83.10%, 74.00%, and 68.90%, corresponding with EBRTs of 56 s, 28 s, and 20 s, respectively.

The variation of RE in mixture was affected not only by ILR, but also by the chemical property of individual contaminant. According to our experiment, the EC of toluene in mixture is enhanced in comparison with the performance as sole treatment in BTF, because of the existing co-metabolism during the mixture contaminant degradation process. Moreover, there are also competitive interactions among different contaminates as Amani has previously indicated [50]. Nevertheless, a competitive effect was not only dependent on the nature of contaminants but also relying on the structure of micro-communities in the environment. Deshusses stated that toluene could be co-treated well with $H_2S$ in order to enhance the performance of BTF [51]. On the contrary, Bentley has reported that VOSCs would have a competitive inhibition by other contaminants during the mixture treatment [23], which was also confirmed by Shu during DMS degradation by BTF system [44]. To conclude, RE was affected by ILR, chemical properties of contaminants, and structure of micro-communities.

### 3.2. Performance of BTF for Single Pollutant Degradation

To compare between degradation of single contaminants and mixtures, each contaminant was tested under different operating parameters (phase I to phase III). The ILR, EC,

and RE of DMS are shown in Figure 4a. Due to the sudden increases in ILR on day 82, 86, 90, and 94, there are several RE drops and EC increases suddenly. The flow rate was increased to 0.40 m$^3$·h$^{-1}$ in order to achieved EBRTs of 28 s in phase II on day 98. ILR was decreased to 13.08 g·m$^{-3}$·h$^{-1}$ from 162.89 g·m$^{-3}$·h$^{-1}$. The similar RE tendency was observed in terms of ILR was suddenly increased. In phase III (day 123 to day 140), the air stream was adjusted to 0.60 m$^3$·h$^{-1}$ to achieve EBRT of 20 s. Difference appeared when the RE was not able to recover by time on day 138, an eliminated capacity was saturated. The maximum EC were observed 141.39 g·m$^{-3}$·h$^{-1}$ (phase I), 115.20 g·m$^{-3}$·h$^{-1}$ (phase II) and 160.54 g·m$^{-3}$·h$^{-1}$ (phase III).

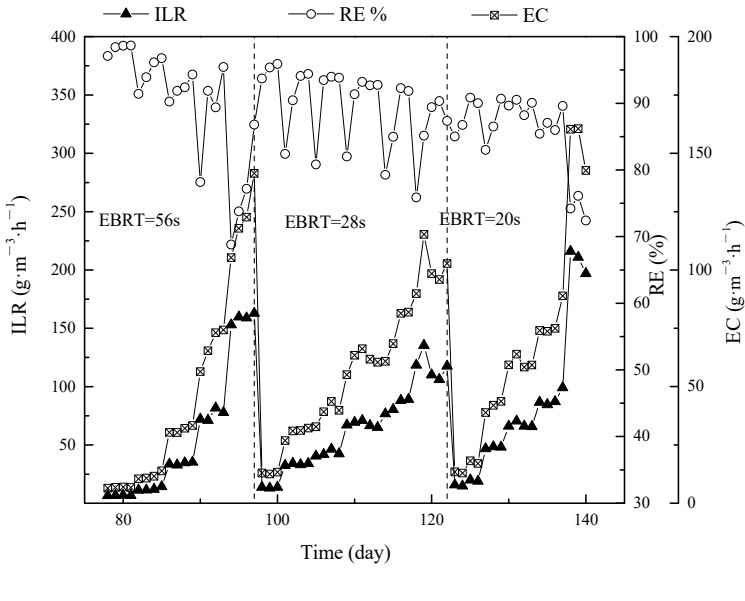

(**a**)

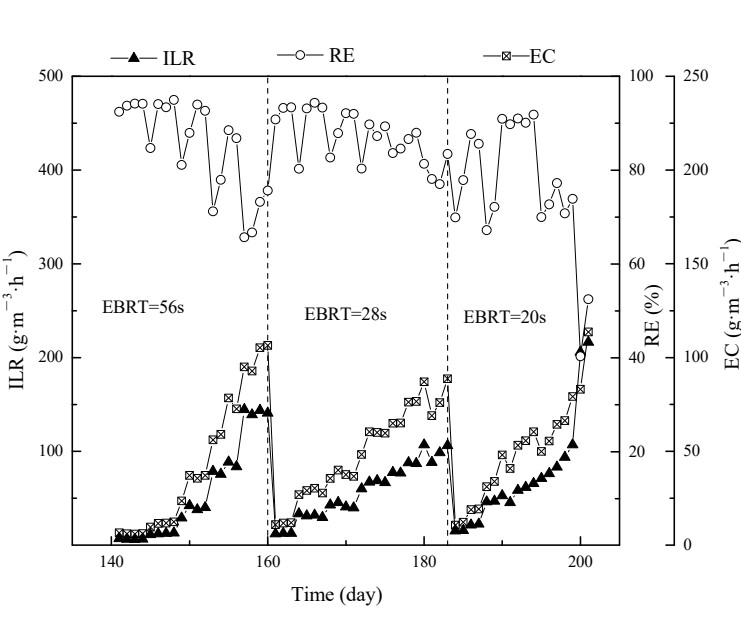

(**b**)

**Figure 4.** *Cont.*

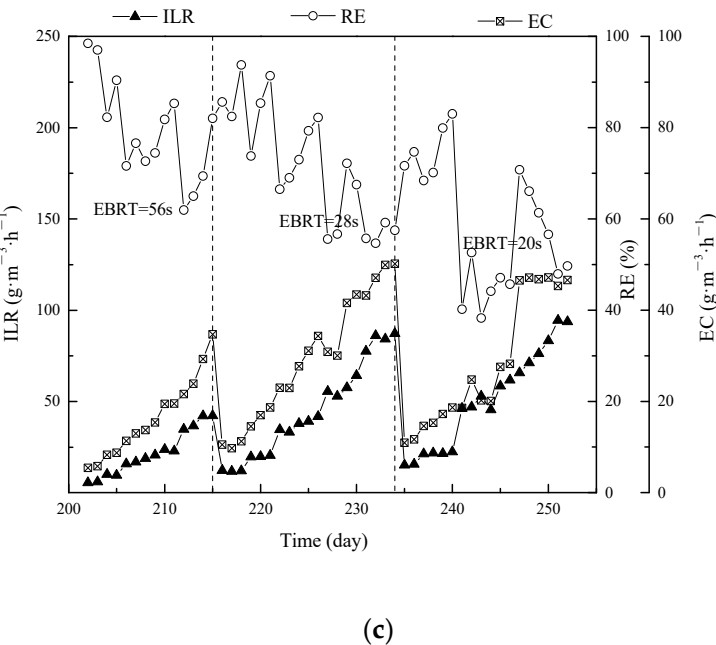

(**c**)

**Figure 4.** Effect of ILR on EC of BTF in single (**a**) DMS; (**b**) propanethiol and (**c**) toluene treatment during different phases.

The description propanethiol and toluene are shown in Figure 4b,c. Compared with EC and RE of propanethiol, toluene was more sensitive with suddenly enhanced ILR, and the variation gaps of EC and RE were amplified. Figures indicated that the BTF was saturated at the end of phase III (day 248), so that continued increasing ILR was not able to enhance EC. On the contrary, an inhibition effect was not negligible in terms of high inlet concentration. The maximum EC of propanethiol and toluene were observed 106.49 $g·m^{-3}·h^{-1}$ and 34.69 $g·m^{-3}·h^{-1}$ (phase I); 88.65 $g·m^{-3}·h^{-1}$ and 50.19 $g·m^{-3}·h^{-1}$ (phase II); 113.64 $g·m^{-3}·h^{-1}$ and 47.17 $g·m^{-3}·h^{-1}$ (phase III). According to Figure 4, the lowest EC was toluene. There are two main reasons: (1) As mentioned in Section 3.5.2, *Betaproteobacteria* (35.61%) and *Gammaproteobacteria* (15.56%) was correlated with DMS and propanehtiol biodegradation. *Sphingobacteriia* (20.54%) and *Alphaproteobacteria* (11.9%) was correlated with toluene biodegradation. (2) The BTF was overall operated 248 days, toluene was exposure up to days 77, and then switched to DMS (days 78–140), propanthiole (days 141–200), finally toluene (days 201–248). Thus there is a starving period 123 days for toluene, it also affected toluene biodegradation performance.

As discussed earlier in Section 3.1, DMS and propanethiol actually performed better in the single treatment process, while toluene was preferred in the ternary treatment rather than single treatment. As shown in Figures 3 and 4, the value of $EC_{max}$ in terms of this study was in order as follows: DMS single > propanethiol single > DMS in mixture > propanethiol in mixture > toluene in mixture > toluene single. This indicated that toluene was the most difficult pollutant to degrade among the three pollutants in either single or mixture system; however, its EC was larger under mixture circumstances revealing that the co-metabolism happened rather than competitive inhibition. Corresponding maximum EC data in previous studies was shown in Table 1.

**Table 1.** Macrokinetic determination of Michaelis–Menten kinetic constant of DMS, propanethiol and toluene in single and mixture treatment under different EBRTs.

| Substrate | $r_{max}$ | $K_s$ | Equation | $R^2$ |
|---|---|---|---|---|
| DMS (single) | | | | |
| EBRT 56 s | 103.09 | 0.40 | Y = 0.00399x + 0.00802 | 0.996 |
| EBRT 28 s | 256.41 | 0.67 | Y = 0.00265x + 0.00375 | 0.999 |
| EBRT 20 s | 232.56 | 0.44 | Y = 0.00259x + 0.00153 | 0.995 |
| Propanethiol (single) | | | | |
| EBRT 56 s | 149.25 | 0.88 | Y = 0.00602x + 0.00511 | 0.997 |
| EBRT 28 s | 204.08 | 0.65 | Y = 0.00319x + 0.00439 | 0.989 |
| EBRT 20 s | 196.08 | 0.53 | Y = 0.00274x + 0.00388 | 0.964 |
| Toluene (single) | | | | |
| EBRT 56 s | 39.84 | 0.22 | Y = 0.00554x + 0.02399 | 0.994 |
| EBRT 28 s | 90.91 | 0.33 | Y = 0.00357x + 0.01099 | 0.990 |
| EBRT 20 s | 82.64 | 0.31 | Y = 0.00385x + 0.00865 | 0.982 |
| DMS (mixture) | | | | |
| EBRT 56 s | 57.80 | 0.28 | Y = 0.00493x + 0.01730 | 0.982 |
| EBRT 28 s | 114.94 | 0.38 | Y = 0.00328x + 0.00917 | 0.999 |
| EBRT 20 s | 97.09 | 0.19 | Y = 0.00510x + 0.00319 | 0.999 |
| Propanethiol (mixture) | | | | |
| EBRT 56 s | 63.69 | 0.34 | Y = 0.00534x + 0.01566 | 0.990 |
| EBRT 28 s | 76.34 | 0.27 | Y = 0.00404x + 0.00713 | 0.980 |
| EBRT 20 s | 104.17 | 0.31 | Y = 0.00296x + 0.00964 | 0.983 |
| Toluene (mixture) | | | | |
| EBRT 56 s | 47.39 | 0.27 | Y = 0.00585x + 0.02111 | 0.985 |
| EBRT 28 s | 99.01 | 0.33 | Y = 0.00430x + 0.00400 | 0.985 |
| EBRT 20 s | 88.50 | 0.21 | Y = 0.00274x + 0.00881 | 0.986 |

*3.3. Carbon Dioxide Production*

In the biofiltration process, the organic contaminants are eventually converted to either water and carbon dioxide or biomass. Therefore, the $CO_2$ production (P $CO_2$) is an essential parameter corresponding with mineralization level. The effect of P $CO_2$ on the EC for various EBRTs is presented in Figure 5. The linear fitting of experiment data under different EBRTs during the mixture gas treatment were y = 2.26x (EBRT of 56 s), y = 1.88x (EBRT of 28 s), and y = 1.72x (EBRT of 20 s). In case of the inlet concentration ratio of DMS, propanethiol and toluene were 2:1:1; therefore, the stoichiometric ratios of P $CO_2$ per mass of completely degradation of mixture contaminants were 2.0, 3.0 and 7.0 respectively. Maximum ECs and P $CO_2$ coefficients of different individual pollutants and mixtures were as shown in Table S2, and among them P $CO_2$ of toluene was studied in various research. Compared to the results in the present study, the P $CO_2$ values in this study were relatively high, in terms of difference between three typical contaminants. The high mineralization rate of mixture contaminants revealed these compounds were entirely converted to water and $CO_2$. The discrepancy between experimental data and stoichiometric values were sensible to explain as an accumulation of biomass in BTFs and partial $CO_2$ was dissolved in the MSM as formed $H_2CO_3$, $HCO_3^{2-}$, or $CO_3^{2-}$, in terms of the degradation process taking place in the aqueous phase.

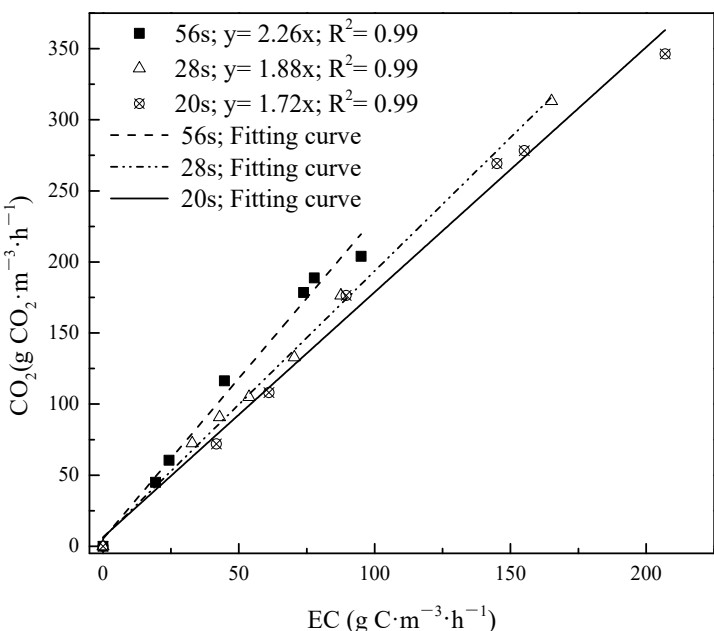

**Figure 5.** Effect of eliminated capacity on $CO_2$ production rate under EBRT of 56 s, 28 s, and 20 s in mixture contaminant BTF treatment during phase I to phase III.

### 3.4. Biodegradation Kinetics for Single and Ternary Mixture Gas

The $r_{max}$ and $K_s$ under different EBRTs for DMS, propanethiol and toluene either in a single or in a mixture gas supply were calculated via Equation (5). As shown in Figure 6a,b, the variation of $r_{max}$ value and $K_s$ corresponded with EBRTs in single and mixture treatment during three phases. For single treatment, $r_{max}$ of DMS, propanethiol and toluene were initially enhanced since EBRT decreased from 56 s to 28 s, and declined with continued decrease of EBRT from 28 s to 20 s. $K_s$ of DMS and toluene followed the $r_{max}$ tendency. However, $K_s$ of propanethiol decreased straight if EBRT dropped. The $K_s$ value indicated that during the single contaminant condition, propanethiol > DMS > toluene. However, during mixture conditions the comparison was complex, under EBRT 56 s, propanethiol > DMS > toluene, under EBRT 28 s DMS > toluene > propanethiol, under EBRT 20 s propanethiol > toluene > DMS. The solubility of contaminants was propanethiol > DMS > toluene. This confirmed that in a single condition, the solubility would positively relate to $K_s$ [52]. But in mixture conditions, the $K_s$ value was not only related with solubility but also related with other facts such as reaction effect, and the intermediates would also influence the results.

In previous studies, Shu calculated that a $r_{max}$ of 115.7 $g·m^{-3}·h^{-1}$ and $K_s$ of 1.56 $g·m^{-3}$ during single DMS biofiltration treatment [53], and Menard reported a $r_{max}$ of 39.4 $g·m^{-3}·h^{-1}$ and $K_s$ of 4.6 $g·m^{-3}$ for toluene degradation [48]. As shown in Table 1, our $r_{max}$ data of DMS (single) and toluene (single) were all apparently higher than previous studies, while $r_{max}$ data of DMS mixture and toluene (mixture) were a bit lower than previous studies. $r_{max}$ value. the maximum amount of DMS degrade per unit of bio-filter volume, estimated from the Michaelis–Menten model was higher than that from the Haldane model (0.63 $gs·gx^{-1}·h^{-1}$), which is consistent with those reported by a previous study [41]. This result also confirmed by other odor gas degradation through BTF system fitting by the Michaelis–Menten model [50].

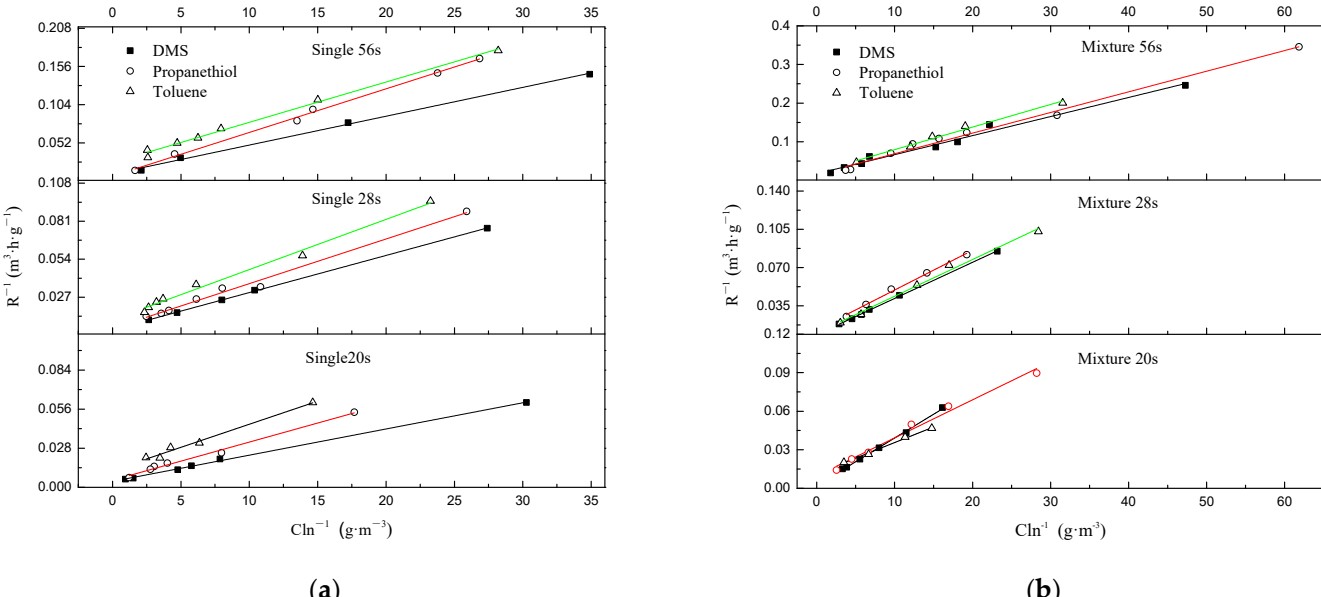

**Figure 6.** Macrokinetic determination of Michaelis–Menten kinetic constant for (**a**) single treatment and (**b**) mixture treatment during different phases.

The value of $r_{max}$ and $K_s$ are shown in Table 1. Compared with single treatment of DMS, either $r_{max}$ or $K_s$ decreased in the mixture treatment of DMS, in terms of the additional carbon resource not only available for microbial communities, it also leads to a reaction effect between different contaminants, resulting in reaction constraints; however, the value of toluene was increased. $r_{max}$ and $K_s$ of DMS and toluene were both initially enhanced from phase I to phase II, and then declined from phase II to phase III. It is possible that the benefit of the intermediates was to enhance the solubility of toluene. These results of $r_{max}$ and $K_s$ are comparable within previous studies of $r_{max}$ of 24.18 g·m$^{-3}$·h$^{-1}$ and 31.10 g·m$^{-3}$·h$^{-1}$, and $K_s$ of 0.62 g·m$^{-3}$ and 0.57 g·m$^{-3}$ for DMS and toluene, respectively [45,54,55].

In general, a sensible inferring of physical identity for $K_s$ is homoplastically to kinetics of enzyme, hence the lower $K_s$ value and the higher enzymatic affinity for contaminants [56]. This explained treatment performance of toluene was better in mixture treatment rather than single treatment. A possible co-metabolism with toluene was the main mechanism during DC area, in terms of its solubility increasing, resulting in the $r_{max}$ enhancement during this stage and $K_s$ decreased under EBRTs 20 s. Co-metabolism/enzymatic affinity is a possible reason why the EC for toluene was enhanced when it was introduced along with the other sulfur-containing compounds. As inlet concentration of mixture gas was continuously increasing and the EBRT was decreasing, co-metabolism plus an occurrence of a reaction effect were both intensified; however, co-metabolism would not compensate the performance loss due to the reaction effect [8,57]. Thus, $r_{max}$ and $K_s$ both declined.

### 3.5. Analysis of Microbial Communities

#### 3.5.1. Bacterial Community Diversity and Composition

Three samples from the BTF system of 20 days, 75 days, and 120 days were analyzed by Illumina high-throughput sequencing, with every length 430 bp and at least 49,515 sequences obtained for each sample. The objective for this comparison was to understand how residence time and the other simulated conditions, such as concentration, components will affect the microbial colonies. The number of OTUs and $\alpha$-diversity of these samples were shown in Table S3. Previously, Ma indicated that larger Shannon index values and smaller Simpson index values were both representative of the higher $\alpha$-diversity [58]. Wang found that bacterial structure became simpler after contaminants degradation [59], which could lead to lower $\alpha$-diversity and OUT number with time passing by the similar trend being observed: the biofilm sampled at 120 days contained the lowest $\alpha$-diversity

(Shannon index 4.08), while the initial biofilm had the highest α-diversity (Shannon index 5.03) and richness (OTU number 2599). After, microbial diversity had been changed. A large number of secondary intermediate products have been produced in the process of microbial degradation of DMS, propyl mercaptan, and toluene [38,41]. A large number of intermediate products generated from the degradation process made the microbial population keep in dynamic equilibrium.

### 3.5.2. Microbial Community Structures at the Phylum Level

In the present study, the microbial analysis was focus on bacteria, and over 99.99% of obtained sequences were assigned bacteria. On average, 3.05% of the sequences were not classified at the phylum level. As shown in Figure 7a, the seven major phyla were *Proteobacteria* (66.42% on average), *Bacteroidetes* (21.76%), *Gemmatimonadetes* (2.00%), *Acidobacteria* (1.89%), *Deinococcus-Thermus* (1.87%), *Firmicutes* (1.38%), and *Candidate* (0.69%). The most abundant phylum was *Proteobacteria*, which ranged from 63.01% to 68.28% in the samples. Among the *Proteobacteria* as shown in Figure 7b, *Betaproteobacteria* ranked first with an average percentage of 35.61%, followed by *Sphingobacteriia* (20.54%), *Gammaproteobacteria* (15.56%) and *Alphaproteobacteria* (11.91%). The high percentage of *Betaproteobacteria* and *Grammaproteobaceria* was as a result of anthropogenic inoculation of *Alcaligenes* sp. SY1 and *Pseudomonas.putia* S1, which was belonged to these classes [23]. *Sphingobacteriia* and *Alphaproteobacteria* were also reported as main class to degrade toluene in bioreactor [60,61]. Thus, these bacteria play key role during VOCs and VOC treatment [62]. As shown in Figure 7b, during the first 20 days operation period, *Gammaproteobacteria* (28.75%) *Sphingobacteriia* (21.57%), *Alphaproteobacteria* (17.63%), *Betaproteobacteria* (13.84%) were the most abundant bacterial class. However, *Betaproteobacteria* would dramatically increase to 42.25%, with the decrease percentage of *Gammaproteobacteria* and *Alphaproteobacteria* to 10.80% and 11.78% for the next 55 days of operation period. At this period, the *Sphingobacteriia* class was slightly declined to 2.00%. In the last 45 days of operation, *Betaproteobacteria* still kept the increased tendency to 50.74%, while *Gammaproteobacteria* and *Alphaproteobacteria* continued declined slightly to 7.12% and 6.32%. *Sphingobacteriia* was increased by 1.50% during this period.

In Figure 7c, *Thiobacillus*, *Pseudomonas*, and *Alcaligenes* were the most abundant genera in this BTF. *Thiobacillus* was dramatically increased from 0.37% (20 days) to 40.24% (75 days) and then slightly declined to 32.72% (120 days). which correlated with toluene in the mixture of VOCs aeration to BTF from day 1 to days 77, then to DMS aeration from days 78 to days 140. *Pseudomonas* was declined from 20.08% (20 days) to 4.67% (75 days) and then recovered to 7.25% (120 days). The drop of *Pseudomonas* is due to *Alcaligenes* inoculation from day 1 to days 20 in startup period, and *Thiobacillus* fast growth from day 1 to day 77 and recovered on day 120 was correlated with DMS aeration. *Alcaligenes* was the most stable genera during three periods, the percentage of *Alcaligenes* was kept from 13.08% (20 days) to 12.09% (75 days) and then reached to 15.55% (120 days), which was under inoculation *Alcaligenes* at startup period and DMS single aeration from days 78 to days 140 are reasonable.

The predominant of *Gammaproteobacteria* during all three periods explained the maximum EC of DMS is relatively higher than the other two contaminants in bottom and middle layer, and also confirmed that *Alcaligenes* sp. SY1 is a promising strain during co-treatment of DMS removal. *Sphingobacteriia* remained constant from the beginning to the end of treatment; consequently, the toluene performance of EC and RE were keep stable in either single or ternary treatment of contaminants in upper layer. The initial deterioration of *Gammaproteobacteria* could be explained the inoculated *Pseudomonas putida*. S1 was affected either by inhibition in the higher inlet load of contaminants or competition effect by other microbial communities. However, the propanethiol performance of EC and RE was not significantly influenced by deterioration due to the ILR was mostly under DC area. The significant enhancement of *Betaproteobacteria* was mainly because of the

growth of *Thiobacillus*, which was reported to apply in VOSCs degradation and toluene degradation [63,64].

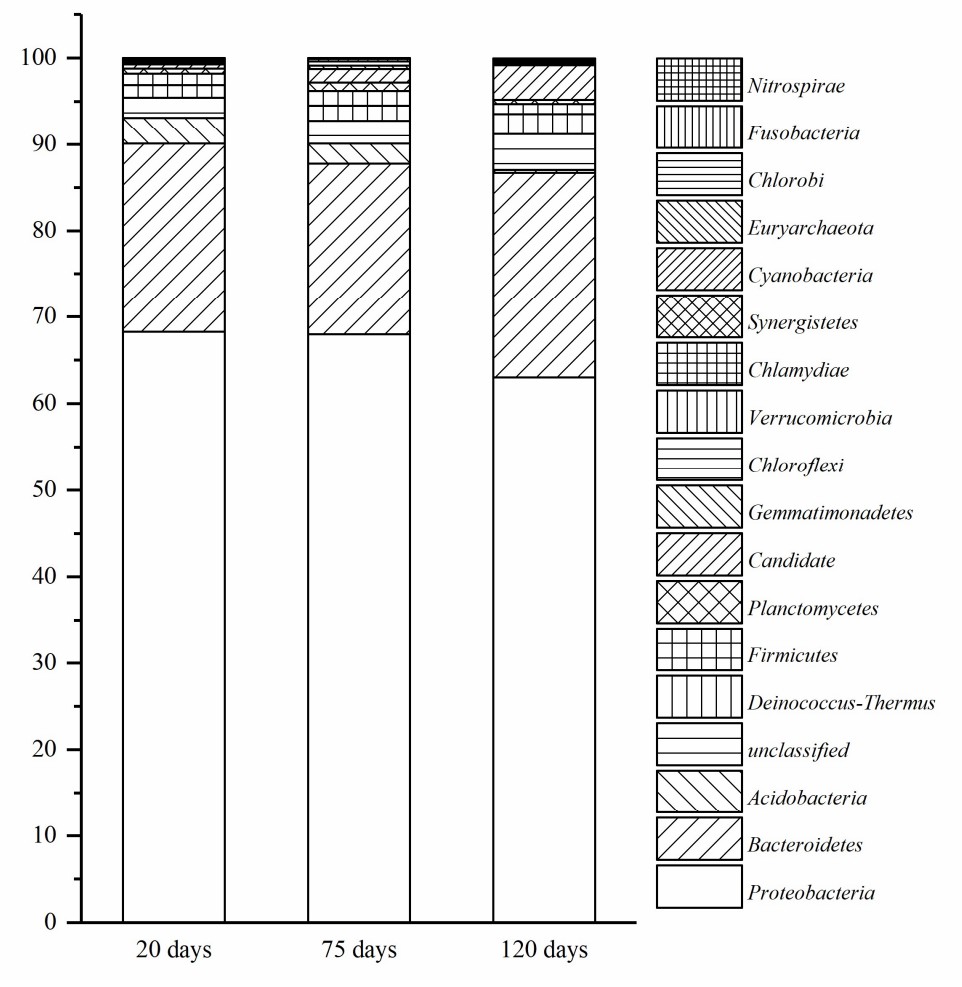

(**a**) Bacterial phyla

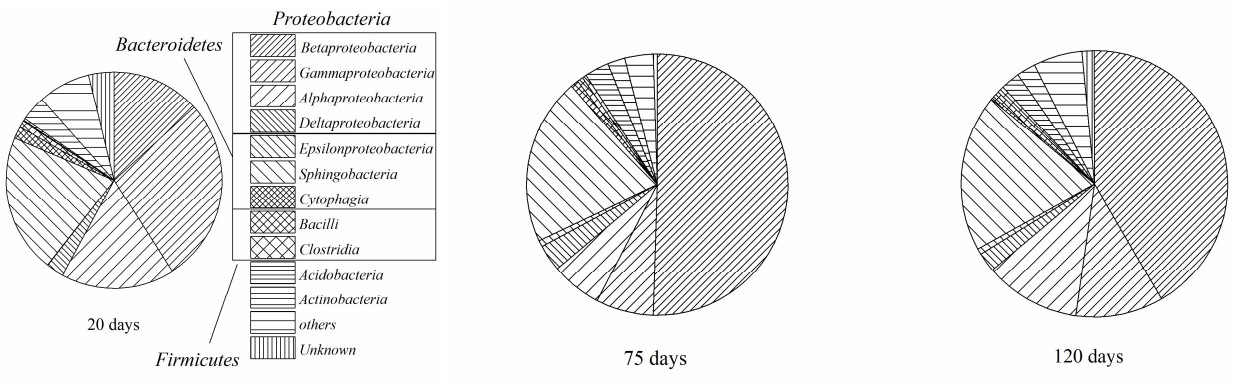

(**b**) Bacterial class

**Figure 7.** *Cont.*

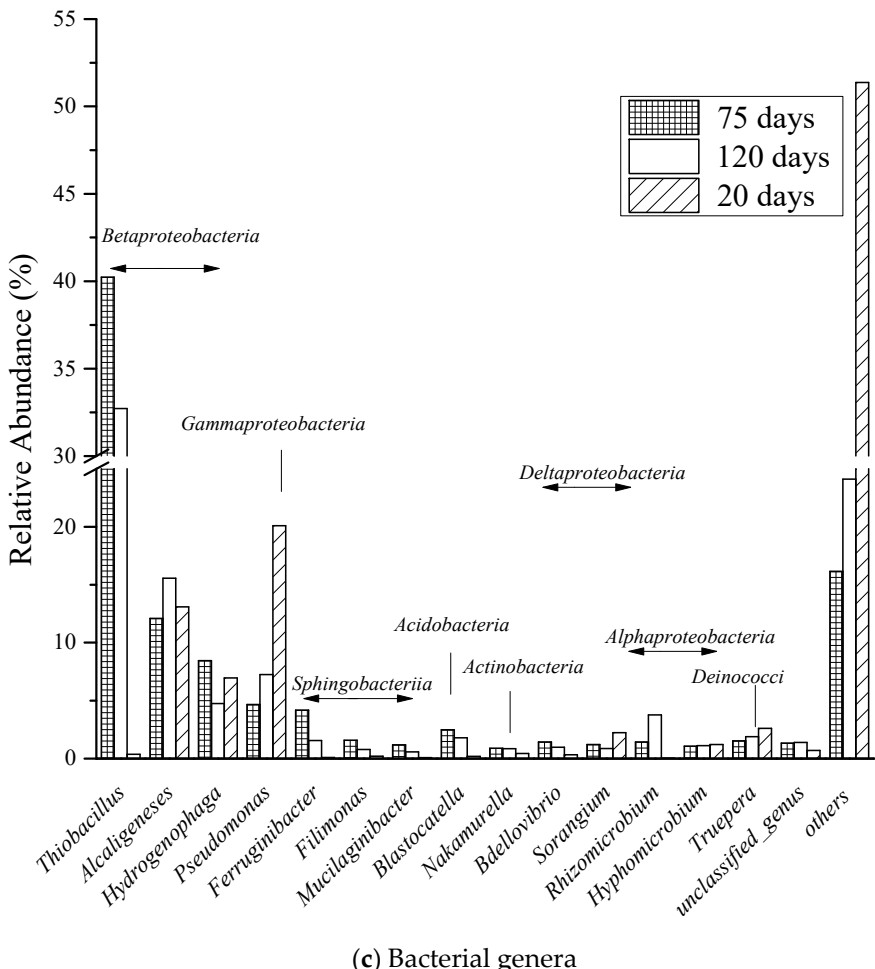

(**c**) Bacterial genera

**Figure 7.** Comparison to quantitative contribution of the sequences affiliated with different (**a**) phyla, (**b**) class, and (**c**) genera to the total number of sequences from microcosm samples at 20 days, 75 days, and 120 days.

### 3.5.3. Microscopic Observations

FESEM was used to observe the biofilm formation and mature as time went by. And The sample for FESEM observation was obtained from the bottom of the first packing layer. As shown in Figure S2a, after 20 days of inoculation, normal density cells number could be observed on the surface of packing material, and between bacterial unit mycelial structures, non-colonized regions on the surface could also be observed. In Figure S2b, high cell growth was observed after 77 days operation, and more than one layer of biofilm adhered on packing material. Globular bacterium and rod-shaped bacterium were predominant in the field. In Figure S2c, the bacterial colonies showed that several layers keep adhesion together with a mycelial structure, could not even distinguish the boundary of a single bacteria unit. The presence of an intensive number of microbial communities was also observed in terms of high density of cell growth. This phenomenon was also observed in toluene and VOCs removal in the BTF system [60,65].

### 4. Conclusions

BTF with *Alcaligenes* sp. SY1 was shown to effectively degrade DMS, propanethiol and, toluene with stable performance. The maximum removal rate ($r_{max}$) for single DMS, propanethiol, and toluene was calculated as 256.41 g·m$^{-3}$·h$^{-1}$, 204.08 g·m$^{-3}$·h$^{-1}$, and 90.91 g·m$^{-3}$·h$^{-1}$, respectively. When they were mixed with the other two, VOC or VOSC, the $r_{max}$ increased for toluene, but decreased for DMS and propanethiol. *Proteobacteria* and *Bacteroidetes* were the major bacterial groups in BTF packing materials, which confirmed that the inoculation did succeed during the BTF process. Therefore, *Alcaligenes* sp. SY1 and

*Pseudomonas putida* S1 has significant potential for treating DMS, propanethiol, and toluene either individually or collectively.

**Supplementary Materials:** The following are available online at https://www.mdpi.com/article/10.3390/fermentation7040309/s1, Figure S1: The schematic of the multi-layer BTF. (1) Air pump, (2,3) rotor meter, (4) sample flask, (5) mixture flask, (6) air inlet, (7) sampling port, (8) Nutrition flask, (9) packing port, (10) spray header, (11) air outlet, (12) packing material, (13) packing material sampling port, (14) air sampling port, (15) pH detector, (16) Auto-pH adjustor, (17) control panel, (18) NaOH solution flask, Figure S2: FESEM microphotographs of the packing material (a); sample of the biofilter on day 20 during startup period (b) sample on day 77 during operational period (c) sample on day 240 during end period, Table S1: Operation condition of BTF treatment, Table S2: Comparison $CO_2$ production of recent results of removal of single or mixed pollutants with present study, Table S3: α-Diversity of each sample.

**Author Contributions:** Y.S. (Yiming Sun), conducted the experiments and wrote the manuscript; X.L., analyzed the results and revised the manuscript; H.L. and S.Z., discussed the results; J.C., designed the experiment and discussed the results; Y.H. and L.Q., participated in the discussion of the manuscript; Y.S. (Yao Shi), designed the experiment and discussed the results. All authors have read and agreed to the published version of the manuscript.

**Funding:** This research was funded by the National Key Research and Development Program of China (grant number 2018YFC0213806), and the National Natural Science Foundation of China (grant number 51308502, 21676245 and 51750110495) for financial support. The work was also sponsored by Key Research and Development Program of HeBei Province (grant number 19273904D), and Postdoctoral Program of China Aerospace KaiTian Environmental Technology (grant number YQB-2017-001). And The APC was funded by Key Research and Development Program of China Aerospace KaiTian Environmental Technology (grant number YQB-2021-001).

**Institutional Review Board Statement:** The study was conducted according to the guidelines of the Ethics Committee of Lincoln University for sensory analysis (Application No: 2020-35, Approved on 14 August 2020).

**Informed Consent Statement:** Informed consent was obtained from all subjects involved in the study.

**Acknowledgments:** The authors would like to thank the National Key Research and Development Program of China (grant number 2018YFC0213806), and the National Natural Science Foundation of China (grant number 51308502, 21676245 and 51750110495) for financial support. The work was also sponsored by Key Research and Development Program of HeBei Province (grant number 19273904D), and Postdoctoral Program of China Aerospace Kaitian Environmental Technology (grant number YQB-2017-001). And the APC was funded by Key Research and Development Program of China Aerospace KaiTian Environmental Technology (grant number YQB-2021-001).

**Conflicts of Interest:** The authors declare no conflict of interest.

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
