# Peer review of "Co-Treatment with Single and Ternary Mixture Gas of Dimethyl Sulfide, Propanethiol, and Toluene by a Macrokinetic Analysis in a Biotrickling Filter Seeded with Alcaligenes sp. SY1 and Pseudomonas Putida S1"

_fermentation, doi:10.3390/fermentation7040309_

Round 1
Reviewer 1 Report
I want to thank the author and co-authors for spending efforts on this interesting work. I am convinced that the author's work is very important in developing technologies to help control environmental pollution, particularly using biotrickling filters for the treatment of waste gases. The authors have laid out logical flow of information in introduction, method, and results combined with discussion.
I skimmed through the manuscript and have noticed some ambiguous presentation/writing, which I listed below. However, this probably does not cover all the parts that should be revised.
- In 2.3 section, I think adding time period details of BTF operations with individual and ternary cotreatment will help prepare the readers to understand the results better. I think the table S1 did not include details of this aspect either. I could be wrong. It looks like the BTF was operated for around 250 days but Table S1 only showed 77 days period.
- In Figure 1, adding the legend for each marker will be helpful for the readers.
- In 2.4, I think there is missing details of sampling for microbial analysis (such as from which days of operations ?) Were samples for FESEM also analyzed for microbial community? But the days in 2.4 not match with those in the Figure 7 in the result section.
- I think narration on the microbial community analysis should be revised to better link with the performance results of BTF. Although the authors already laid out some context on this perspectives, I think there is some missing pieces. For example, I am not sure is there the explaination why samples on 20, 75, and 120 days were selected? And was there correlation of microbial community result to the BTF operations around the same period that samples were collected? Any observation that indicated the co-metabolism for toluene degradation?
- In the conclusion section, I suggest the author revise it to have consistency with the overall manuscript. But again, I could be wrong as I skimmed through the manuscript. I think the BTF was inoculated with not only the two identified isolates but also the acclimatized activated sludge. I think read only at the conclusion, I would be misunderstood.
- I suggest the author make another few times of language check. I am not qualified to assess the language but I noticed some strange writings. Please correct me if I am wrong. In line 367, "An possible co-metabolism" why "An" was used here?
In overall, I think this work showed a great research effort but needs some more editions to better help the reader understanding.
Hope you find this helpful,
Thank you again for your hard work
Regards
Author Response
Response to Reviewer #1
Q1:In 2.3 section, I think adding time period details of BTF operations with individual and ternary cotreatment will help prepare the readers to understand the results better.(1)
I think the table S1 did not include details of this aspect either. I could be wrong. It looks like the BTF was operated for around 250 days but Table S1 only showed 77 days period.(2)
Response: (1) We appreciate the reviewer comments and have corrected the issue in the revised manuscript.
(2) The BTF was overall operated 252 days, and the table S1 indicated parameters changes such as ILR and EBRT from startup period as phase I to days 77 as phase IV. we have corrected the issue in the revised manuscript.
Q2:In Figure 1, adding the legend for each marker will be helpful for the readers.
Response: We appreciate the reviewer comments and have corrected the issue in the revised manuscript as legend in Figure 1.
Q3:In 2.4, I think there is missing details of sampling for microbial analysis (such as from which days of operations ?) Were samples for FESEM also analyzed for microbial community? (1)
But the days in 2.4 not match with those in the Figure 7 in the result section. (2)
Response: (1)The sample for FESEM observation was taken from the bottom of first packing layer on days 20 of startup period, on days 77 of steady period, on days 240 of end period to shown microbial morphological characteristics of different period of BTF operation. samples taken from days 20, 75 and 120 was for analyzed for microbial community to indicate that microbial activity under different ILR and EBRT.
(2) Figure 7 was shown the microbial community develop from days 20 to days 120 from startup period to steady period in molecular level under different ILR and EBRT. in Fig. S2 was shown microbial morphological characteristics from overall operation cycle as startup period, steady period and end period. the purpose of two parts experiment was designed different.
Q4:I think narration on the microbial community analysis should be revised to better link with the performance results of BTF. Although the authors already laid out some context on this perspective, I think there is some missing pieces. For example, I am not sure is there the explaination why samples on 20, 75, and 120 days were selected? (1)
And was there correlation of microbial community result to the BTF operations around the same period that samples were collected? Any observation that indicated the co-metabolism for toluene degradation? (2)
Response: (1) We appreciate the reviewer comments. The objective for this comparison is to understand how residence time and the other simulated conditions such as concentration, components will affect the microbial colony. We have clarified this reason in the revised manuscript.
(2) The overall EC of contaminants were followed the order as: DMS single > propanethiol single > DMS in mixture > propanethiol in mixture > toluene in mixture > toluene single. the DMS EC was mainly correlated with Gammaproteobacteria due to Pseudomonas putida. S1. the toluene was mainly correlated with Betaproteobacteria because of the growth of Thiobacillus, and it was due to co-metabolism that EC of toluene mixture was higher than toluene single.
Q5:In the conclusion section, I suggest the author revise it to have consistency with the overall manuscript. But again, I could be wrong as I skimmed through the manuscript. I think the BTF was inoculated with not only the two identified isolates but also the acclimatized activated sludge. I think read only at the conclusion, I would be misunderstood.
Response: Following to our previous research, the biofilm culturing of Alcaligenes sp.SY1 was growth fast when the ratio of mixture compounds was set up to 2:1:1 under laboratory condition. and it is correct, few acclimatized activated sludges were used in system. we have revised in 2.1.
Q6:I suggest the author make another few times of language check. I am not qualified to assess the language but I noticed some strange writings. Please correct me if I am wrong. In line 367, "An possible co-metabolism" why "An" was used here?
Response: We have significantly revised the manuscript according to the review’s suggestions.
Reviewer 2 Report
Comments to the manuscript fermentation-1461372: Co-treatment with single and ternary mixture gas of dimethyl sulfide, propanethiol, and toluene by microkinetic analysis in a biotrickling filter seeded with Alcaligenes sp. SY1 and Pseudomonas putida S1
This manuscript deals with the degradation of DMS, propanethiol, and toluene under single and ternary conditions and the correlation with the present microflora. The topic is interesting and analytical data show interesting aspects. However, the manuscript in the current form shows weaknesses in the linguistic presentation, in the literature work, and especially in the critical evaluation of the own results. Detailed comments are as followed:
- Abstract Line 20: air pollution caused by volatile organic compounds
- Line 37: VOSCs in general show poor water solubility.
- Line 51: Please delete ‘filtration’ as VOSCs and VOCs are not particle based. Dilution is also not a method to remove the compounds. Please add absorption and as UV is listed as traditional approach non-thermal plasma should be added, too. Adequate references might be:
- https://www.osti.gov/servlets/purl/15009785
- https://doi.org/10.1002/jctb.5515
- https://doi.org/10.1016/j.jece.2017.10.015
- https://doi.org/10.3390/su12208577
- In general: Please have a look on large letters all over the text. In a couple of situations small letters are required.
- Line 60: Incomplete sentence.
- Line 62: the chlorine is corrosive against metal.
- Line 63: No. 3) Please revise this sentence. By the way, the disadvantages of RTO and RCO are listed here. Hence, listing BTFs is not adequate.
- Line 64: intermediates
- Line 65: Compared
- Line 66: As BTFs generally convert VOCs to CO2 they are commonly no swamp for CO2, but will form CO2 as mineralization product.
- Line 69: biotrickling filters
- Line 69-70: most probably ‘field’. In general the distinction of BTF and bioscrubbers in not correct. Bioscrubbers operate at higher water loads and with water soluble VOCs while BTF operate at lower water loads or even intermittent irrigation and for mixtures of VOCs or commonly more lipophilic VOCs.
- Line 70: The waste water amount of BTFs is very low. Bioscrubbers are suitable for centralized waste water treatment scenarios.
- Line 76: References 18 and 19 only focus on sulfur containing compounds. Please add references for chlorine or nitrogen containing compounds. Adequate references are:
- https://doi.org/10.1007/s00253-011-3255-x
- https://doi.org/10.1016/j.bej.2008.09.007
- https://doi.org/10.1016/j.scitotenv.2018.05.278
- https://doi.org/10.1016/j.ejbt.2018.04.004
- https://doi.org/10.3303/cet1654051
- Line 78: There are a couple of papers dealing with ethanethiol. For example:
- https://doi.org/10.1007/s11270-012-1126-4
- https://doi.org/10.1007/s10532-010-9366-8
- https://doi.org/10.1155/2015/414237
- https://doi.org/1080/09593330.2018.1545804
- https://doi.org/10.1016/j.jhazmat.2010.07.035
- https://doi.org/10.1080/10962247.2013.763305
- https://doi.org/10.1186/s12896-019-0540-8
- Please change this statement
- Line 80: There are also a couple of papers dealing with DMDS in biotrickling filters
- doi: 10.1080/09593330902911713
- DOI: 10.2166/wst.2012.365
- https://doi.org/10.1016/j.biortech.2014.11.002
- https://doi.org/10.1016/j.biortech.2015.11.081
- DOI:1016/j.cej.2007.11.038
- Line 81 and line 86: Why thioanisole?? According to the header it is toluene. Please clarify.
- Line 93: References 18 and 19 do not focus on toluene. Another reference should be added.
- Line 95-98: There are a couple of more papers dealing with VOSCs and VOCs in mixture. Please change your sentences.
- https://doi.org/10.1016/j.tibtech.2018.02.004
- Line 121: The figure of the BTF is a core aspect of this work and should be shown within the main document.
- Line 186: The core operational parameter should be listed in the main text.
- Figure 1: What are the conditions of the startup phase (EBRT?). A legend should be implemented.
- Line 223: …’while no significant outlet enhancement was observed during phase I and phase II’…. What do you mean with that statement. Clean gas levels increased in all three phases.
- General: Please avoid presentation of concentrations and efficiencies with two digits after the point. One is enough as there are limitations in the detection.
- Line 224-225: Not clear. There is an increase during phase III.
- Line 226: Abbreviation ILR should be introduced.
- Line 236-237: Statement is critical as the gap increases with reduced EBRT.
- Line 246: For this statement the solubility of the three compounds should be critically addressed. à also relevant for Line 250 and statement of 253.
- Line 257-258: Delete this sentence as the statement is absolutely clear.
- Line 281: …not able to be recovered by time…
- Line 282: an eliminated capacity was saturated. What do you mean with that?
- General statement: There are a couple of lingual aspects affecting the quality of presentation. Please have a detailed look about grammar.
- Line 284: The description propanthiol and toluene…. What do you mean with that?
- Line 292-300: Please try to explain why toluene showed lowest EC. Most probably the inoculum and the start up period are the core aspects for the explanation. According to Figure 4 it seems that way that there was toluene exposure up to day 70 (first three periods) and then a change of VOCs to DMS (day 70-140), propanthiole (140-200) and finally to toluene. Hence, there is a gap in toluene supply for 130 days or more, affecting for sure the degradability of toluene according to a lack in toluene degraders. According to the inoculum. Are these strains able to degrade toluene?
- Figure 4c: Scale for RE should start at 0.
- Figure 5: Commonly EC is given as g C/(m³*h). Most probably the unit of CO2 is g CO2/(m³*h). It should be addressed as g C/(m³*h) to see a clear correlation.
- Line 343: The presentation of rmax is okay, but as conversion strongly depends on diffusive mass transport which is strongly affected by the type of package material and irrigation management, additional data should be presented here and critically compared to the current data.
- Line 374: reaction
- Line 289: These intermediates were not addressed before. Do you have additional informations (concentration of elementary sulfur or sulfate, benzoic acid…)
- Chapter 3.5.2: Three samples were taken after days 20, 75 and 120. According to the figures this is the time after phase 0 (start-up), at the end of the feed of ternary mixture (75) and during the DMS feeding (120). According to the higher loadings of sulfuric compounds it seems to be clear that Pseudomonas sp., most probably able to degrade toluene, declined in ratio. However, the interesting part (that is a sample during the single feed of toluene and also propanthiol) is not shown. Please add these data and try a critical correlation/explanation considering degradation potentials of these three bacterial species listed in Line 416.
Author Response
Response to Reviewer #2
Dear Reviewer 2:
Q1:Abstract Line 20: air pollution caused by volatile organic compounds (1)
Line 37: VOSCs in general show poor water solubility.(2)
Response: (1)(2) We appreciate the reviewer comments and have corrected the issue in the revised manuscript.
Q2:Line 51: Please delete ‘filtration’ as VOSCs and VOCs are not particle based. Dilution is also not a method to remove the compounds. Please add absorption and as UV is listed as traditional approach non-thermal plasma should be added, too. Adequate references might be:
https://www.osti.gov/servlets/purl/15009785
https://doi.org/10.1002/jctb.5515
https://doi.org/10.1016/j.jece.2017.10.015
https://doi.org/10.3390/su12208577
Response: We appreciate the reviewer comments and have changes in reference.
Q3:In general: Please have a look on large letters all over the text. In a couple of situations small letters are required.(1)
Line 60: Incomplete sentence.
Line 62: the chlorine is corrosive against metal.
Line 63: No. 3) Please revise this sentence. By the way, the disadvantages of RTO and RCO are listed here. Hence, listing BTFs is not adequate.
Line 64: intermediates
Line 65: Compared
Line 66: As BTFs generally convert VOCs to CO2 they are commonly no swamp for CO2, but will form CO2 as mineralization product.
Line 69: biotrickling filters(2)
Response: (1)(2) We have corrected these mistakes and carefully proofread the revised manuscript.
Q3:Line 70: The waste water amount of BTFs is very low. Bioscrubbers are suitable for centralized waste water treatment scenarios.(1)
Line 76: References 18 and 19 only focus on sulfur containing compounds. Please add references for chlorine or nitrogen containing compounds. Adequate references are(2)
https://doi.org/10.1007/s00253-011-3255-x
https://doi.org/10.1016/j.bej.2008.09.007
https://doi.org/10.1016/j.scitotenv.2018.05.278
https://doi.org/10.1016/j.ejbt.2018.04.004
https://doi.org/10.3303/cet1654051
Line 78: There are a couple of papers dealing with ethanethiol. For example:
https://doi.org/10.1007/s11270-012-1126-4
https://doi.org/10.1007/s10532-010-9366-8
https://doi.org/10.1155/2015/414237
https://doi.org/1080/09593330.2018.1545804
https://doi.org/10.1016/j.jhazmat.2010.07.035
https://doi.org/10.1080/10962247.2013.763305
https://doi.org/10.1186/s12896-019-0540-8
Line 80: There are also a couple of papers dealing with DMDS in biotrickling filters(3)
doi: 10.1080/09593330902911713
DOI: 10.2166/wst.2012.365
https://doi.org/10.1016/j.biortech.2014.11.002
https://doi.org/10.1016/j.biortech.2015.11.081
DOI:1016/j.cej.2007.11.038
Response: (1) We appreciate the reviewer comments and have revised manuscript.
(2) We appreciate the reviewer comments and have changes in reference
(3) We have made changes in manuscript.
Q4:Line 81 and line 86: Why thioanisole?? According to the header it is toluene. Please clarify.(1)
Line 93: References 18 and 19 do not focus on toluene. Another reference should be added.
Line 95-98: There are a couple of more papers dealing with VOSCs and VOCs in mixture. Please change your sentences.(2)
https://doi.org/10.1016/j.tibtech.2018.02.004
Line 121: The figure of the BTF is a core aspect of this work and should be shown within the main document.
Line 186: The core operational parameter should be listed in the main text.
Figure 1: What are the conditions of the startup phase (EBRT?). A legend should be implemented.
Line 223: …’while no significant outlet enhancement was observed during phase I and phase II’…. What do you mean with that statement. Clean gas levels increased in all three phases.
General: Please avoid presentation of concentrations and efficiencies with two digits after the point. One is enough as there are limitations in the detection.
Line 224-225: Not clear. There is an increase during phase III.
Line 226: Abbreviation ILR should be introduced.(3)
Response: (1) it is an example of VOSCs component for biologically treatment
(2) We appreciate the reviewer comments and have changes in reference
(3) We appreciate the reviewer comments and have revised manuscript.
Q5:General: Please avoid presentation of concentrations and efficiencies with two digits after the point. One is enough as there are limitations in the detection.(1)
Line 224-225: Not clear. There is an increase during phase III.
Line 226: Abbreviation ILR should be introduced.(2)
Line 236-237: Statement is critical as the gap increases with reduced EBRT.(3)
Line 246: For this statement the solubility of the three compounds should be critically addressed. à also relevant for Line 250 and statement of 253.
Line 257-258: Delete this sentence as the statement is absolutely clear.
Line 281: …not able to be recovered by time…
Line 282: an eliminated capacity was saturated. What do you mean with that?
General statement: There are a couple of lingual aspects affecting the quality of presentation. Please have a detailed look about grammar.
Line 284: The description propanthiol and toluene…. What do you mean with that?(4)
Response:(1)(2)(4) We appreciate the reviewer comments and have revised manuscript.
(3) EC was increased with reduced EBRT under ILR increased condition.
Q6:General statement: There are a couple of lingual aspects affecting the quality of presentation. Please have a detailed look about grammar (1)
Line 284: The description propanthiol and toluene…. What do you mean with that? (2)
Response: (1) (2)We have corrected these mistakes and carefully proofread the revised manuscript.
Q7:Line 292-300: Please try to explain why toluene showed lowest EC. Most probably the inoculum and the start up period are the core aspects for the explanation.
Response: We appreciate the reviewer comments, the main aspects for toluene showed lowest EC was depending on inoculum is correct, as mentioned in 3.5.2, Betaproteobacteria (35.61%) and Gammaproteobacteria (15.56%) was correlated with DMS and propanethiol biodegradation. Sphingobacteriia (20.54%) and Alphaproteobacteria(11.9%) was correlated with toluene biodegradation.
Q8:According to Figure 4 it seems that way that there was toluene exposure up to day 70 (first three periods) and then a change of VOCs to DMS (day 70-140), propanthiole (140-200) and finally to toluene. Hence, there is a gap in toluene supply for 130 days or more, affecting for sure the degradability of toluene according to a lack in toluene degraders. According to the inoculum. Are these strains able to degrade toluene?
Response: We appreciate the reviewer comments, the operation condition is correct, the experiment was designed not only for testing ILR and EBRT effect the performance of strains but also nesteia period and recovered time for strains in further studies. The nesteia period in experiment is same, and lacking performance for toluene is correlated with strains adaptation for all experiment parameters. Not all strains are suitable for toluene degradation, the main strain were Sphingobacteriia (20.54%) and Alphaproteobacteria(11.9%) as mentioned in 3.5.2.
Q9:Figure 4c: Scale for RE should start at 0.(1)
Figure 5: Commonly EC is given as g C/(m³*h). Most probably the unit of CO2 is g CO2/(m³*h). It should be addressed as g C/(m³*h) to see a clear correlation.(2)
Response: (1) We appreciate the reviewer comments, scale for RE should start at 0 normally, but RE line would cross with ILR line and EC line. it might be difficult for reader to observe.
(2) We appreciate the reviewer comments, this comparison method was cited from Ref.36 (Álvarez-Hornos, F.J.; Gabaldón, C.; Martínez-Soria, V.; Marzal, P.; Penya-roja, J.M. Mathematical modeling of the biofiltration of ethyl acetate and toluene and their mixture. Biochem. Eng. J. 2009, 43, 169-177.) Mathematical modeling of the biofiltration of ethyl acetate and toluene and their mixture to understand relationship between elimination capacity (EC) and production of carbon dioxide.
Q10:Line 343: The presentation of rmax is okay, but as conversion strongly depends on diffusive mass transport which is strongly affected by the type of package material and irrigation management, additional data should be presented here and critically compared to the current data.
Response: We appreciate the reviewer comments, it is correct that rmax is depending on few key parameters such as diffusive mass, packing material and irrigation. in previous studies, packing material and irrigation is quite different. thus, list of comparison might aware the difference, but difficult to have solid conclusion.
Q11:Line 374: reaction (1)
Line 389: These intermediates were not addressed before. Do you have additional informations (concentration of elementary sulfur or sulfate, benzoic acid…)(2)
Response: (1) We have corrected this mistake.
(2) We appreciate the reviewer comments, in previous studies, the intermediates of DMS, propanethiol are all reported in details, we have revised and cited as Ref.40. Ref.43 in this part.
Q12:Chapter 3.5.2: Three samples were taken after days 20, 75 and 120. According to the figures this is the time after phase 0 (start-up), at the end of the feed of ternary mixture (75) and during the DMS feeding (120). According to the higher loadings of sulfuric compounds it seems to be clear that Pseudomonas sp., most probably able to degrade toluene, declined in ratio. However, the interesting part (that is a sample during the single feed of toluene and also propanthiol) is not shown. Please add these data and try a critical correlation/explanation considering degradation potentials of these three bacterial species listed in Line 416.
Response: We appreciate the reviewer comments, the objective for this comparison is to understand how residence time and the other simulated conditions such as concentration, components will affect the microbial colony especially of Alcaligenes continue with previous study, we have clarified this correlation of BTF operation condition and microbial colony shifted in the revised manuscript.
Reviewer 3 Report
This work investigates the utilization of BTF for the treatment of a mixture of VOSC and VOCs. In general, the merit of the study is there. However, the study is written in poor language with numerous grammatical mistakes. There are no legend available in most of the figures. Moreover, there is no convincing arguments why these particular gases were chosen at the influent. The time of operation of each phase is very limited. There is a declining trend suggesting that no steady state was achieved in all phases.
The following are some of the comments
- VOSCs are abbreviated twice differently in the first paragraph
- Line 44: Need a reference showing the average ranges of influent in the pesticide industry
- Line 44: DMS was listed without abbreviation. It was only abbreviated later in line 47
- Line 53: the references 5 and 6 reflect the utilization of biological treatment not physio-chemical techniques
- Line 58: need references showing the constituents of waste gas
- Line 60: statement starts with small letter
- Line 62: Do not start statements with numbers
- Line 80: Which studies look for DMDS?
- Figure 1: No legend present
Author Response
Response to Reviewer #3
Dear Reviewer 3:
Q1:VOSCs are abbreviated twice differently in the first paragraph
Response: We have please provided more evidence why biotrickling filter is irreplaceable in this area and better than conventional approaches in the revised manuscript. We have revised line 54-62 so that the introduction is more concise than the previous manuscript.
Q2:Line 44: Need a reference showing the average ranges of influent in the pesticide industry (1)
Line 44: DMS was listed without abbreviation. It was only abbreviated later in line 47 (2)
Response: (1) the ranges of concentration was not from reference but from real industry in field detected by online FID facility combine with manual monitoring by lab.
(2) We appreciate the reviewer comments and have corrected the issue in the revised manuscript in line 42
Q3:Line 53: the references 5 and 6 reflect the utilization of biological treatment not physio-chemical techniques (1)
Line 58: need references showing the constituents of waste gas (2)
Response: We appreciate the reviewer’s comments and revised in the manuscript.
Q4:Line 60: statement starts with small letter (1)
Line 62: Do not start statements with numbers (2)
Line 80: Which studies look for DMDS? (3)
Figure 1: No legend present (4)
Response: (1)(2)(3) We have corrected these mistakes and carefully proofread the revised manuscript
(4) we have revised Figure 1.
Round 2
Reviewer 2 Report
Comments to the manuscript fermentation-1461372-R1: Co-treatment with single and ternary mixture gas of dimethyl sulfide, propanethiol, and toluene by microkinetic analysis in a biotrickling filter seeded with Alcaligenes sp. SY1 and Pseudomonas putida S1
This manuscript was dramatically improved in quality. Most aspects were addressed. Finally, three aspects are still not taken into consideration and should be addressed:
- Line 294-301: Please try to explain why toluene showed lowest EC. Most probably the inoculum and the start up period are the core aspects for the explanation. According to Figure 4 it seems that way that there was toluene exposure up to day 70 (first three periods) and then a change of VOCs to DMS (day 70-140), propanthiole (140-200) and finally to toluene. Hence, there is a gap in toluene supply for 130 days or more, affecting for sure the degradability of toluene according to a lack in toluene degraders. According to the inoculum. Are these strains able to degrade toluene?
- Figure 4c: Scale for RE should start at 0.
- Figure 5: Commonly EC is given as g C/(m³*h). Most probably the unit of CO2 is g CO2/(m³*h). It should be addressed as g C/(m³*h) to see a clear correlation.